# Omni-fMRI: A Universal Atlas-Free fMRI Foundation Model

**Mo Wang** [* 1 2 3]  **Wenhao Ye** [* 1 4]  **Junfeng Xia** [1]  **Junxiang Zhang** [5 1]  **Xuanye Pan** [1]  **Minghao Xu** [3]  **Haotian Deng** [1]
**Hongkai Wen** [3]  **Quanying Liu** [1 2 5]

## Abstract

Self-supervised fMRI foundation models have shown promising transfer performance, yet most rely on predefined region-level parcellations that discard fine-grained voxel information and introduce atlas-dependent biases. We propose Omni-fMRI, an atlas-free foundation model that operates directly on voxel-level signals. To enable scalable pretraining on 49,497 fMRI sessions across nine datasets, Omni-fMRI introduces a dynamic patching mechanism that substantially reduces computational cost while preserving informative spatial structure. To support reproducibility and fair comparison, we establish a comprehensive benchmark suite spanning 11 datasets and a diverse set of resting-state and task-based fMRI tasks. Experimental results demonstrate that Omni-fMRI consistently outperforms existing foundation models, providing a scalable and reproducible framework for atlas-free brain representation learning. Code is available at Link.

## 1. Introduction

Functional Magnetic Resonance Imaging (fMRI) is a cornerstone of non-invasive neuroimaging, providing millimeter-scale measurements of Blood-Oxygen-Level-Dependent (BOLD) signals across the entire brain. Modern scanners produce dense 4D spatiotemporal volumes comprising millions of voxels, posing substantial challenges for representation learning. To reduce dimensionality, a dominant paradigm aggregates voxel-wise signals into Regions of Interest (ROIs) defined by predefined parcellations. This

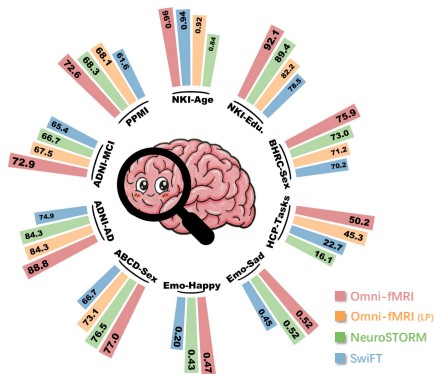

*Figure 1. Omni-fMRI* reaches the best performance across a diverse array of resting-state and task-based fMRI benchmarks.

abstraction yields region-level time series amenable to Convolutional Neural Networks (Kawahara et al., 2017) or enables the construction of functional connectivity graphs for Graph Neural Networks (Li et al., 2021). While effective for specific supervised tasks, such end-to-end pipelines tend to learn representations that exhibit limited out-of-distribution generalization (Kawahara et al., 2017; Li et al., 2021; Ye et al., 2023). To mitigate this limitation, recent efforts have shifted toward task-agnostic fMRI foundation models trained via self-supervised learning (SSL) on large-scale unlabeled cohorts. These models have demonstrated improved robustness and transferability across downstream tasks (Dong et al., 2024; 2025; Yang et al., 2024; Caro et al., 2023; Wang et al., 2025c; Wei et al., 2026; Xia et al., 2026).

However, despite these architectural advances, current fMRI foundation models largely inherit the same input representation paradigm as earlier supervised approaches, relying heavily on predefined parcellations. The paradigm for fMRI input representation has not undergone a comparable shift. Existing approaches continue to rely on predefined parcellations that project high-dimensional 4D volumes into low-dimensional regional summaries. While such projections facilitate the adoption of standard computer vision or graph-based architectures, they inevitably compromise information fidelity by discarding fine-grained voxel-level signals. More critically, commonly used atlases are derived from group-level aggregation and therefore obscure substantial inter-subject functional variability, introducing systematic

---
[*]Equal contribution  [1]Department of Biomedical Engineering, Southern University of Science and Technology, China [2]Omni-Intelligence, China [3]Department of Computer Science, University of Warwick, The UK [4]School of Biomedical Engineering, Shenzhen, China [5]Center for Language, Intelligence and Machines, Shenzhen Loop Area Institute, Shenzhen, China. Correspondence to: Quanying Liu < liuqy@sustech.edu.cn >.

*Proceedings of the 43rd International Conference on Machine Learning*, Seoul, South Korea. PMLR 306, 2026. Copyright 2026 by the author(s).

misalignment across individuals. Although subject-specific parcellations could, in principle, alleviate this heterogeneity, scaling individualized optimization to the tens of thousands of scans required for foundation model pretraining is computationally prohibitive and incompatible with large-scale SSL pipelines. Furthermore, since no single atlas is universally optimal across tasks or populations, the choice of parcellation introduces an additional source of inductive bias that directly impacts downstream performance (Wang et al., 2025b; Salehi et al., 2020). The absence of a standardized and atlas-independent input representation consequently hinders fair benchmarking, transferability, and model reuse.

To address these limitations, we propose *Omni-fMRI*, an atlas-free fMRI foundation model that operates directly on voxel-level signals. By eliminating reliance on predefined parcellations, Omni-fMRI avoids atlas-induced misalignment and preserves fine-grained functional information. Inspired by adaptive patch strategies in computer vision (Rao et al., 2021; Havtorn et al., 2023; Choudhury et al., 2025), we introduce a dynamic patching mechanism to render voxel-level modeling computationally tractable at scale. This approach assigns larger patches to background or spatially redundant regions, while preserving fine-grained resolution in functionally salient areas. This data-driven tokenization substantially reduces memory and computational overhead without sacrificing representational fidelity. Addressing the challenge of inconsistent dataset versions and evaluation protocols that hinder fair comparison in the fMRI field, we establish a benchmark spanning 11 datasets and 16 different downstream tasks. To ensure strict reproducibility and transparency, we release full experiment logs and exact test subject IDs, enabling standardized evaluation across diverse cognitive, demographic, and clinical prediction settings. Extensive empirical results demonstrate that Omni-fMRI consistently outperforms existing state-of-the-art fMRI foundation models across various resting-state and task-based fMRI benchmarks (Fig. 1). Our contributions are summarized as follows:

- We propose **Omni-fMRI**, an atlas-free voxel-level fMRI foundation model with a **dynamic patching strategy** that enables training substantially larger models under reduced wall-clock time.

- We demonstrate **state-of-the-art performance** across a wide range of downstream tasks, and show that linear probing with Omni-fMRI outperforms the fully fine-tuned results of several existing fMRI models.

- We establish a **comprehensive and reproducible benchmark** for fMRI representation learning, providing standardized datasets, evaluation protocols, and released experiment logs to facilitate fair comparison and future research.

## 2. Related Work

While several foundation models for fMRI have been proposed, the predominant approach relies on ROI-based or graph-based inputs (Dong et al., 2024; 2025; Yang et al., 2024). One reason is that applying the most popular vanilla Vision Transformers (ViTs) directly to patch-level 4D fMRI is computationally impractical (Dosovitskiy et al., 2020). Global self-attention scales quadratically with the number of tokens, making direct ViT-based modeling infeasible in both memory and computation.

To address this challenge, hierarchical Transformer architectures have been widely adopted for dense visual inputs, including fMRI. Representative designs such as Swin restrict self-attention to local windows and cyclically shift window partitions across layers, enabling efficient modeling of high-dimensional inputs on regular grids (Fan et al., 2021; Liu et al., 2021). Recent voxel-level fMRI models predominantly employ this strategy, as shown in Table 1. NeuroSTORM represents the most closely related contemporary fMRI foundation model within this paradigm. It employs a Mamba-based backbone and adopts the shifted-window attention and hierarchical patch merging strategy. This hierarchical design substantially reduces the computational burden of voxel-level modeling and enables large-scale pretraining.

However, such hierarchical designs impose structural constraints particularly detrimental to fMRI analysis. First, restricting attention to local windows compromises the model's ability to capture long-range functional dependencies, which are ubiquitous in brain organization. Consequently, establishing critical global connectivity requires indirect, multi-layer message passing—an inefficient process given the brain's extensive long-range networks (Eickhoff et al., 2018). Second, the reliance on fixed, dense window partitioning binds the backbone to a regular lattice, hindering the effective exclusion of signal-free background regions. Given that non-brain background typically constitutes over 50% of a 4D fMRI volume, this leads to significant computational redundancy. Third, a unified patch merging strategy

*Table 1.* **Supervised and Self-Supervised Voxel-level fMRI Models.** "Reduction" refers to the downsampling strategy. The symbol * indicates pretrained foundation model.

| Model | Main Tasks | Reduction |
|---|---|---|
| Omni-fMRI* (Ours) | General-purpose | Dynamic Patch |
| NeuroSTORM* (Wang et al., 2025a) | General-purpose | Shifted Windows |
| SLIM-Brain* (Wang et al., 2025c) | Diagnosis | Mask Unit |
| Doodipala et al. (2025) | Diagnosis | Shifted Windows |
| DCA (Wang et al., 2025b) | Atlas Generation | Shifted Windows |
| Sun et al. (2025) | State Prediction | Shifted Windows |
| SwiFUN (Kwon et al., 2024) | State Prediction | Shifted Windows |
| SWIFT (Kim et al., 2023) | Demography | Shifted Windows |
| Shi et al. (2023) | State Prediction | 3D CNN |
| TFF (Malkiel et al., 2021) | Demography | 3D CNN |

**a. Masked Autoencoding with Dynamic Patching**

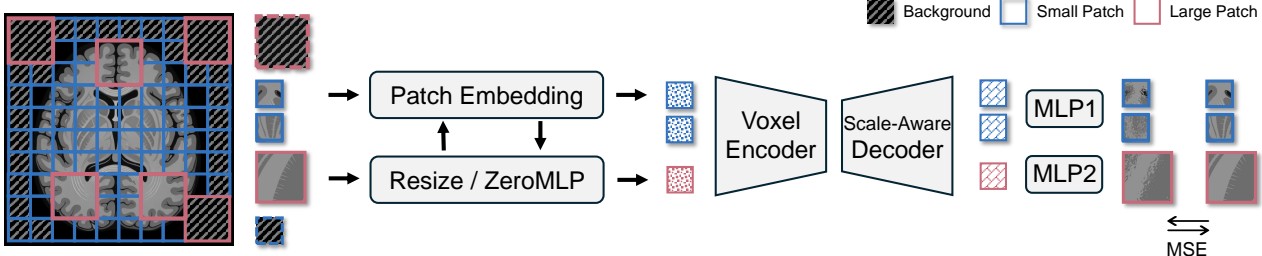

**b. Content-Adaptive Patch Allocation**     **c. Dual-Path Multi-Scale Embedding**     **d. Scale-Aware Reconstruction**

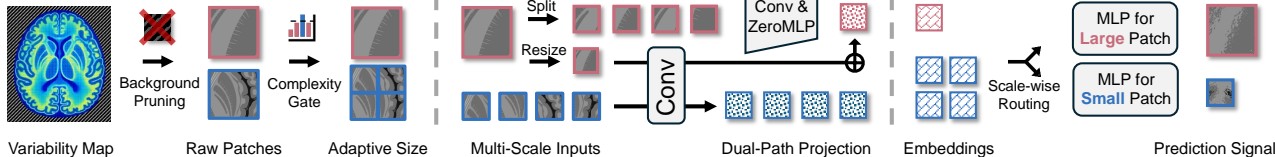

*Figure 2.* **Overview of the Omni-fMRI Pre-training Framework. (a)** The proposed masked autoencoder architecture leverages a dynamic patching strategy to process 4D fMRI volumes. By filtering out background noise and adapting patch resolution, the model achieves computational efficiency while preserving fine-grained details. **(b) Content-Adaptive Patch Allocation.** A spatiotemporal complexity map is computed to guide tokenization: non-informative regions are pruned (Background Pruning), while foreground regions are dynamically assigned coarse (Red outline) or fine (Blue outline) patches based on local signal variability via a Complexity Gate. **(c) Dual-Path Multi-Scale Embedding.** To align heterogeneous patches into a unified latent space, we employ a dual-path projection module. Fine patches are projected directly, while coarser patches utilize a residual branch with Zero-initialized MLP (ZeroMLP) to fuse high-frequency structural details. **(d) Scale-Aware Reconstruction.** The reconstruction routes latent tokens to scale-specific prediction heads, utilizing distinct MLPs to reconstruct voxels at their appropriate resolution for high-fidelity restoration.

can lead to excessive downsampling, compromising informative signals in highly active brain regions. In contrast, we adopt a dynamic patching strategy. By selectively merging tokens in simpler areas and discarding background while preserving detail in complex ones, we drastically reduce the total token sequence length (from approximately 14 K to 4.3 K). This efficiency allows our model to leverage a standard ViT architecture with global self-attention, effectively capturing long-range activities across the entire brain.

## 3. Methods

### 3.1. Problem Formulation and Overview

We study self-supervised pretraining of foundation models directly on voxel-wise resting-state fMRI. A 4D fMRI volume is denoted as $\mathbf{X} \in \mathbb{R}^{H \times W \times D \times T}$, where $H, W, D$ are spatial dimensions and $T$ is the number of time points. Our objective is to learn a task-agnostic encoder $f_\theta : \mathbf{X} \to \mathbf{Z}$ that captures intrinsic spatiotemporal brain dynamics without relying on predefined atlases, ROI parcellations, or region-level aggregation.

From a representation learning perspective, voxel-wise fMRI constitutes an *irregular spatiotemporal field* with highly heterogeneous information density. In particular, large portions of the volume correspond to non-informative background, while foreground tissue exhibits strong local

redundancy, with spatially adjacent voxels often showing highly similar activity patterns. This violates the implicit assumption of uniform information density underlying standard grid-based tokenization in vision transformers.

We address this mismatch by jointly designing the tokenization, embedding, and self-supervised objective (Fig. 2): (i) a dynamic patch tokenization mechanism that adapts spatial granularity to local spatiotemporal complexity, (ii) a dual-path multi-scale embedding that stabilizes optimization under heterogeneous patch resolutions, and (iii) a scale-aware masked autoencoding objective that enables principled self-supervised learning across mixed-granularity tokens.

### 3.2. Dynamic Patch Tokenization for Voxel-wise fMRI

**Limitations of Uniform Tokenization.** Uniform partitioning of a $96^3$ fMRI volume with a patch size of $4^3$ yields over 13,000 tokens per time frame, rendering standard global self-attention computationally infeasible. More fundamentally, uniform tokenization allocates identical capacity to background voxels and information-dense tissue, resulting in both inefficiency and representational dilution.

Rather than enforcing hierarchical or window-based structures that still require dense token grids, we adopt a *dynamic patching strategy* that explicitly models the non-uniform information distribution of fMRI volumes (Fig. 2b). This

design allows us to employ a standard ViT encoder and MAE pretraining framework, while feeding only informative, non-background tokens into the transformer.

**Spatiotemporal Complexity Estimation.** To guide adaptive patch partitioning, we estimate local spatiotemporal complexity using time-aggregated intensity variance. For a local patch volume $\mathcal{P}$, we define the complexity score as

$$\sigma_{\mathcal{P}}^2 = \frac{1}{T} \sum_{t=1}^{T} \left( \mathbb{E}_{\mathcal{P}}[I_t^2] - (\mathbb{E}_{\mathcal{P}}[I_t])^2 \right), \qquad (1)$$

where $I_t$ denotes voxel intensity at time $t$, and $\mathbb{E}_{\mathcal{P}}[\cdot]$ is implemented via efficient 3D average pooling. Unlike spectral or entropy-based alternatives, variance is scale-consistent and non-parametric. Moreover, unstructured thermal noise in fMRI exhibits low spatial coherence and is largely suppressed by spatial averaging within $\mathcal{P}$, such that $\sigma_{\mathcal{P}}^2$ reflects the structural richness of the patch, ensuring that more tokens are allocated to regions with higher informative content. Ablations are in Appendix C.2.

**Background Removal and Patch Selection.** A defining characteristic of fMRI volumes is the presence of large regions that are structurally guaranteed to be non-informative. Unlike window-based transformers, our formulation does not require maintaining a fixed window shape. We therefore explicitly remove background patch if its mean intensity value falls below a predefined threshold.

For the remaining foreground regions, patch granularity is determined via a coarse-to-fine strategy. Patches with low complexity ($\sigma_{\mathcal{P}}^2 < \tau$) are represented using a single coarse token (red outline patch in Fig. 2b), while regions exceeding the given threshold $\tau$ are recursively subdivided into finer sub-patches as base size (blue outline patch).

**Dual-Path Multi-Scale Embedding.** Dynamic patching yields tokens with heterogeneous spatial resolutions, which must be projected into a shared latent space before transformer encoding. To this end, we employ a dual-path multi-scale embedding module (Fig. 2c), adapted from adaptive patch tokenization frameworks (Choudhury et al., 2025).

Patches at the base resolution are embedded directly using a 3D convolutional tokenizer. For a larger patch $P_{\text{large}}$, we construct a composite embedding:

$$\mathbf{z} = \phi(P_{\downarrow}) + \text{ZeroMLP}(\text{Conv}(\phi(P_{\text{grid}}))) + \mathbf{p}_{\text{pos}}, \quad (2)$$

where $\phi$ denotes the patch tokenization module implemented via 3D convolution. $P_{\downarrow}$ is a downsampled representation of $P_{\text{large}}$, capturing its low-frequency contextual information. $P_{\text{grid}}$ denotes a grid of non-overlapping sub-patches extracted from $P_{\text{large}}$. The term $\mathbf{p}_{\text{pos}}$ denotes the positional embedding associated with the patch.

The aggregation operator $\text{Conv}$, implemented as a strided convolution, fuses sub-patch features to recover structural details that are lost during downsampling. The residual pathway is modulated by a ZeroMLP whose output is initialized to zero, such that the model initially relies exclusively on the low-frequency embedding $\phi(P_{\downarrow})$. This design induces a curriculum-like optimization behavior, allowing high-frequency spatial details to be incorporated gradually as training progresses.

### 3.3. Scale-Aware Masked Autoencoding

We adopt the masked autoencoder (MAE) paradigm for self-supervised pretraining. However, standard MAE implicitly assumes a uniform voxel budget per token, an assumption that breaks down under heterogeneous patching. Applying a standard MAE objective under heterogeneous patching leads to a scale-biased reconstruction loss, where large patches dominate optimization irrespective of semantic importance. We therefore introduce a scale-aware decoding and reconstruction objective.

**Scale-Conditioned Decoding.** Let $\mathbf{h}_i \in \mathbf{Z}$ denote the latent representation of token $i$ produced by the encoder. To explicitly encode spatial granularity, we inject a learnable scale embedding into the decoder input:

$$\mathbf{u}_i = \mathbf{h}_i + \mathbf{p}_i + \mathbf{e}_{s_i}, \qquad (3)$$

where $\mathbf{p}_i$ is the positional embedding, $\mathbf{e} \in \mathbb{R}^{K \times C}$ is a learnable scale embedding table, and $s_i \in \{0, \ldots, K-1\}$ denotes the scale index, $C$ is the dimension.

**Multi-Scale Reconstruction Heads.** Because patches at different scales correspond to vastly different voxel volumes, larger patches would otherwise dominate the training signal. As a result, a single shared reconstruction head is insufficient. We therefore employ a bank of scale-specific predictors $\{\psi_s\}_{s=0}^{K-1}$. Each predictor maps a latent token to a flattened voxel vector $\hat{\mathbf{y}} \in \mathbb{R}^{T \cdot V_s}$, where $V_s$ denotes the spatial volume of scale $s$ (Fig. 2d).

Let $\mathcal{M}$ denote the set of masked tokens and let $\mathcal{M}_s$ be the subset of masked tokens at scale $s$. The pretraining objective is defined as:

$$\mathcal{L} = \sum_{s=0}^{K-1} \frac{1}{|\mathcal{M}_s| \cdot V_s} \sum_{i \in \mathcal{M}_s} \|\psi_s(\text{Decoder}(\mathbf{u}_i)) - \mathbf{y}_i\|_2^2, \tag{4}$$

where $\mathbf{y}_i$ denotes the ground-truth voxel values. By normalizing the error by both token frequency ($|\mathcal{M}_s|$) and patch volume ($V_s$), this formulation yields a scale-invariant signal, preventing larger patches from dominating the optimization.

*Table 2.* Summary of the **datasets** used in this study. The pretraining data was curated to cover a diverse spectrum of age groups. Age is presented as mean ± standard deviation unless otherwise specified. Please refer to Appendix for details.

| ID | Dataset | Participants | Sessions | Age | Sex (M/F) | Downstream Tasks |
|----|---------|-------------|----------|-----|-----------|------------------|
| **Type I: Only for Pre-training** | | | | | | |
| 1 | UK Biobank (Sudlow et al., 2015) | 38,301 | 38,372 | $68.37 \pm 7.48$ | 17879/20493 | - |
| 2 | AOMIC (PIOP1) (Snoek et al., 2021) | 168 | 672 | $22.15 \pm 1.83$ | 94/67 | - |
| 3 | AOMIC (PIOP2) (Snoek et al., 2021) | 180 | 720 | $21.99 \pm 1.83$ | 98/81 | - |
| 4 | CHCP (Ge et al., 2023) | 244 | 1,224 | 18-79 (Range) | 117/127 | - |
| 5 | ISYB (Gao et al., 2022) | 130 | 520 | $22.52 \pm 2.61$ | 98/32 | - |
| **Type II: Internal Downstream Tasks** | | | | | | |
| 6 | ABCD (Casey et al., 2018) | 1,680 | 6,720 | $9.93 \pm 0.63$ | 813/866 | Sex Classification |
| 7 | ABIDE (Di Martino et al., 2014) | 609 | 2,436 | 6-58 (Range) | 506/103 | Age Regression |
| 8 | HCP rest (Van Essen et al., 2013) | 606 | 2,424 | $28.79 \pm 3.53$ | 321/281 | Sex Classification |
| 9 | PPMI (Marek et al., 2011) | 331 | 1,324 | 35-84 (Range) | 181/150 | Disease Classification |
| **Type III: External Downstream Tasks** | | | | | | |
| 10 | ADNI (Jack Jr et al., 2008) | 497 | 1,988 | 55-90 (Range) | - | Disease Classification |
| 11 | SALD (Wei et al., 2017) | 493 | 493 | 19-80 (Range) | 197/306 | Age Regression |
| 12 | BHRC (de Oliveira et al., 2024) | 465 | 465 | $9.93 \pm 1.79$ | 272/193 | Sex Classification |
| 13 | NKI (Telesford et al., 2023) | 717 | 717 | 7-86 (Range) | 311/406 | Education/Age regression |
| 14 | NSD (Allen et al., 2021) | 6 | 70,566 | 19-32 (Range) | - | Image Retrieval |
| 15 | HCP task (Van Essen et al., 2013) | 118 | 1,647 | 23-36(Range) | 54/64 | State Prediction |
| 16 | StudyForrest (Hanke et al., 2016) | 20 | 71,980 | 26.6 (Mean) | 12/8 | Emotion Prediction |

## 4. Results

### 4.1. Datasets and Parameters

As detailed in Table 2, we utilize a diverse collection of public neuroimaging datasets categorized into three distinct groups. Type I consists of five datasets reserved only for pre-training. Type II comprises four datasets, partitioned with 70% allocated to pre-training and 30% to internal downstream tasks. Finally, Type III encompasses seven datasets designated solely for external downstream evaluation. All images were resampled to a resolution of 2 mm isotropic using cubic B-spline interpolation, followed by padding or cropping to fixed spatial dimensions of $(H, W, D) = (96, 96, 96)$. For datasets with differing temporal resolutions, we further interpolated the time series to a uniform sampling rate of 0.72 s TR, also using cubic B-spline interpolation. Voxel-level data were normalized using global Z-scoring, while data for baseline methods was processed according to their respective specifications.

Omni-fMRI was pretrained with a batch size of 24 on four NVIDIA A10G GPUs (24GB) for 35 epochs, requiring approximately 32 hours. For downstream tasks, Omni-fMRI was fine-tuned for up to 30 epochs. We conducted necessary hyperparameter searches for the baseline models and trained them to convergence. We use a CLS token as the sequence-level representation, with a one-layer MLP head for downstream classification, regression, and CLIP embedding alignment tasks. Task descriptions (Appendix A.1) and the baselines (Appendix A.2), along with configuration

details for our model (Appendix A.3) are provided.

### 4.2. Downstream Tasks

**Demographic Tasks** As shown in Table 3, we fine-tune and evaluate our model across 9 datasets covering a range of demographic tasks, including age regression, as well as sex, disease status, and education level classification. We report effect sizes (Cohen's d) rather than relying solely on significance tests, given the number of runs (n=3) (Cohen, 2013). Across all evaluations, our model outperforms most baselines, yielding accuracy improvements of up to 10 percentage points and indicating that it learns domain-invariant representations that transfer effectively to unseen datasets and tasks.

**Image Retrieval** We evaluate image retrieval performance using the NSD. Models are fine-tuned on 515 shared images viewed by 6 subjects. Decoding performance is quantified using a standard 100-way image retrieval task. We report Top-1, Top-5, and Top-10 accuracy metrics in Tab. 4. Our model outperforms the baseline foundation model. It should be noted that, because the NSD paradigm uses rapid event-related visual stimulation, the hemodynamic responses to adjacent stimuli are temporally overlapping. We therefore use GLM-estimated single-trial responses rather than raw BOLD time series for this analysis. This experiment should be viewed primarily as a test of whether a resting-state-pretrained model can generalize to GLM-derived stimulus-evoked responses (Appendix B.1).

*Table 3.* Performance on demography and disease diagnosis comparison with Accuracy/F1 and MSE/Pearson correlation. Red indicates the best performance and Underline indicates the second performance. * denotes a large effect size with Cohen's d $\geq$ 0.8.

| Model \ Dataset | ABIDE *Age Regression* | | NKI *Age Regression* | | SALD *Age Regression* | | ABCD *Sex Classif.* | | HCP *Sex Classif.* | |
|---|---|---|---|---|---|---|---|---|---|---|
| | MSE ↓ | R ↑ | MSE ↓ | R ↑ | MSE ↓ | R ↑ | ACC ↑ | F1 ↑ | ACC ↑ | F1 ↑ |
| Brain-LM | 0.907±0.007 | 0.239±0.023 | 0.503±0.015 | 0.678±0.014 | 0.677±0.055 | 0.634±0.061 | 59.24±1.03 | 58.74±0.70 | 62.71±4.43 | 61.74±4.72 |
| Brain-JEPA | 0.982±0.066 | 0.166±0.017 | 1.030±0.080 | 0.330±0.030 | 1.130±0.108 | 0.299±0.060 | - | - | 69.97±2.73 | 69.17±3.55 |
| BrainMass | 0.695±0.038 | 0.500±0.044 | 0.600±0.062 | 0.620±0.036 | 0.704±0.080 | 0.626±0.062 | 57.92±1.19 | 57.86±1.18 | 67.65±1.02 | 66.93±1.78 |
| SwiFT | 0.524±0.010 | 0.704±0.011 | 0.127±0.004 | 0.946±0.002 | 0.164±0.08* | 0.926±0.003* | 66.67±3.51 | 68.69±0.44 | 90.75±4.38 | 90.76±4.37 |
| NeuroSTORM | 0.537±0.016 | 0.648±0.013 | 0.152±0.033 | 0.835±0.035 | 0.194±0.019 | 0.892±0.010 | 76.51±1.88 | 76.24±1.70 | 94.17±2.65 | 94.17±2.15 |
| **Omni-fMRI** | **0.427±0.006*** | **0.734±0.002*** | **0.088±0.004*** | **0.959±0.003*** | 0.182±0.019 | 0.909±0.005 | 77.01±0.55 | 76.91±0.54 | 95.54±0.36 | 95.05±0.41 |

| Model \ Dataset | BHRC *Sex Classif.* | | PPMI *PD Diagnosis* | | ADNI (MCI) *Diagnosis* | | ADNI (AD) *Diagnosis* | | NKI *Education Classif.* | |
|---|---|---|---|---|---|---|---|---|---|---|
| | ACC ↑ | F1 ↑ | ACC ↑ | F1 ↑ | ACC ↑ | F1 ↑ | ACC ↑ | F1 ↑ | ACC ↑ | F1 ↑ |
| Brain-LM | 57.09±1.33 | 54.69±1.03 | 54.86±1.20 | 49.03±2.64 | 64.53±2.01 | 61.35±0.66 | 80.96±0.46 | 72.43±0.64 | 60.46±0.73 | 58.93±0.70 |
| Brain-JEPA | 58.16±6.10 | 56.16±6.05 | 47.91±3.61 | 48.60±2.42 | 64.53±0.60 | 55.17±0.66 | 80.23±0.80 | 73.26±1.03 | 49.21±2.15 | 48.73±1.62 |
| BrainMass | 62.76±1.50 | 61.05±1.22 | 58.68±4.21 | 51.95±2.23 | 62.39±0.60 | 61.35±0.60 | 73.45±3.91 | 62.10±0.70 | 62.36±1.24 | 61.98±0.88 |
| SwiFT | 70.21±0.02 | 69.56±0.02 | 61.55±2.52 | 58.23±4.20 | 65.38±1.29 | 72.54±0.16 | 74.90±1.88 | 80.76±0.33 | 78.54±4.87 | 79.08±3.85 |
| NeuroSTORM | 73.02±2.19 | 72.49±1.96 | 68.25±1.18 | 65.18±2.55 | 66.67±1.06 | 62.17±8.93 | 84.26±0.80 | 83.91±0.75 | 89.35±4.72 | 89.34±4.49 |
| **Omni-fMRI** | **75.86±1.35*** | **75.96±1.31*** | **72.57±0.98*** | **70.19±0.84*** | **72.92±1.64*** | **72.87±1.74** | **88.77±0.71*** | **88.60±0.72*** | **92.05±2.85** | **91.64±2.25** |

*Table 4.* Comparison of **image retrieval** performance (Accuracy %) averaged over three runs. "Top-1", "Top-5", and "Top-10" denote the accuracy metrics in a 100-way retrieval setting.

| Model | Top-10 | Top-5 | Top-1 |
|---|---|---|---|
| TFF | 6.5 ± 0.71 | 3.67 ± 0.47 | 0.33 ± 0.24 |
| SwiFT | 10.57 ± 1.40 | 5.67 ± 1.89 | 1.65 ± 0.85 |
| NeuroSTORM | 20.83 ± 1.65 | 11.67 ± 0.47 | 3.00 ± 0.71 |
| Omni-fMRI | **23.08 ± 1.93** | **14.12 ± 1.08** | **6.93 ± 0.48** |

**Emotion Detection** We evaluated the model's performance on emotion detection using the StudyForrest dataset (Hanke et al., 2016). The fMRI time series were segmented into blocks and labeled using the dataset's average emotion scores. Given the highly unbalanced distribution of emotion labels, we formulated the task as a regression problem, specifically focusing on the frequently occurring emotions of *happiness* and *sadness*. To ensure robustness, experiments were conducted across three random data splits. We benchmarked our approach against recent voxel-level models, including NeuroSTORM and SwiFT. As shown in Table 5, our model consistently outperforms these foundation models. Detailed experimental designs are provided in Appendix B.2.

*Table 5.* **Emotion Detection**. Bold indicates the best. * denotes a large effect size with Cohen's d $\geq$ 0.8.

| Model | Happiness MSE ↓ / R ↑ | Sadness MSE ↓ / R ↑ |
|---|---|---|
| SwiFT | 0.849 ± 0.063 / 0.430 ± 0.025 | 0.928 ± 0.104 / 0.449 ± 0.055 |
| NeuroSTORM | 1.410 ± 0.070 / 0.201 ± 0.062 | 1.412 ± 0.184 / 0.520 ± 0.059 |
| Omni-fMRI | **0.459 ± 0.222*** / **0.472 ± 0.013*** | **0.691 ± 0.117*** / **0.521 ± 0.077** |

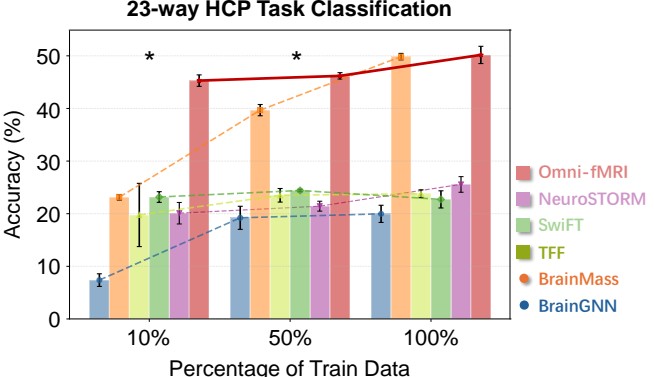

*Figure 3.* **23-way HCP Task Accuracy under different few-shot levels.** * denotes a large effect size with Cohen's d $\geq$ 0.8.

**State Detection** We evaluated the model's performance on brain state prediction using the HCP Task dataset. fMRI time series were segmented into task activation blocks based on experimental triggers, formulated as a 23-way classification problem. To ensure robustness and fair comparison, experiments were conducted across three random data splits, with identical data partitions used for all models. We further assessed data efficiency by benchmarking Omni-fMRI against state-of-the-art baselines (BrainMASS, SwiFT, TFF, BrainGNN, NeuroSTORM) under full supervision (100%) and few-shot settings (50%, 10%). As shown in Fig. 3, the results demonstrate remarkable performance consistency across varying data regimes. Notably, Omni-fMRI exhibits superior data efficiency, maintaining high accuracy even with only 10% of the training data, whereas baselines suffer significant performance degradation. Details are in Appendix B.3.

*Table 6.* Linear Probing performance on **demography and disease diagnosis** comparison with Accuracy/F1 and MSE/R metrics. **Red** indicates the best performance. * denotes a large effect size with Cohen's d ≥ 0.8.

| Dataset
Model | PPMI
*PD Diagnosis* | | ADNI (AD)
*Diagnosis* | | NKI
*Education Classif.* | | NKI
*Age Regression* | | SALD
*Age Regression* | |
|---|---|---|---|---|---|---|---|---|---|---|
| | ACC ↑ | F1 ↑ | ACC ↑ | F1 ↑ | ACC ↑ | F1 ↑ | MSE ↓ | R ↑ | MSE ↓ | R ↑ |
| SwiFT | 61.55±2.52 | 58.23±4.20 | 74.69±2.86 | 74.70±2.63 | 64.29±0.91 | 63.47±0.91 | 0.713±0.016 | 0.654±0.050 | 0.933± 0.042 | 0.460±0.131 |
| NeuroSTORM | 66.78±0.51 | 57.37±0.86 | 71.64±0.66 | 65.35±2.06 | 79.63±4.58 | 80.39±4.55 | 0.304±0.012 | 0.829±0.007 | 0.386±0.035 | 0.774 ± 0.004 |
| **Omni-fMRI** | **68.13±0.60*** | **63.65±2.04*** | **84.26±0.87*** | **83.65±1.37*** | **82.18±1.31** | **80.98±1.60** | **0.162±0.012*** | **0.919±0.003*** | **0.354±0.048*** | **0.814±0.024*** |

## 4.3. Linear Probing

To assess the intrinsic quality of the learned representations, we employ a linear probing protocol. By freezing the pretraining encoder and training a lightweight linear classifier, linear probing (LP) provides a direct measure of the linear separability of the latent features and their ability to capture task-relevant information (He et al., 2022). We evaluate LP on various downstream tasks as shown in Table 6 and Fig. 4. Results show that Omni-fMRI consistently outperforms baseline encoders and, in certain instances, even surpasses their fine-tuned counterparts. Notably, our method exhibits markedly lower performance degradation in the linear probing setting compared to NeuroSTORM (3% versus 13%).

## 4.4. Ablation

We conduct comprehensive ablation studies to evaluate key components of our framework, including scaling (Appendix C.1), complexity metrics (Appendix C.2), patch normalization (Appendix C.3), self-supervised proxy tasks (Appendix C.4), mask ratio (Appendix C.5) and complexity thresholds.

**Scaling Laws** We analyze performance scaling along two orthogonal axes: model size and data volume. For model scaling, we pretrained two architectural variants with different backbone sizes on a subset of 3,000 sessions. For data scaling, we fixed the architecture to the Base configuration and varied the pretraining corpus size from 3,000 to 50,000 sessions. We evaluate the performance on 4 downstream tasks, including 2 internal tasks and 2 external generalization tasks. As shown in Fig. 4, downstream performance improves consistently with increases in both model size and data volume. More tasks and details are in Appendix C.1.

**Ablation of Patch Partitioning** To assess the robustness of our dynamic tokenization strategy, we benchmark the proposed local intensity variance metric against three alternative complexity criteria: Local Shannon Entropy, Laplacian Response, and MSE. Evaluated via end-to-end training on the HCP and ADNI datasets, our selected metric demonstrates a superior efficiency-performance trade-off (Table 7). Further details are provided in Appendix C.2.

*Table 7.* Comparison of different methods for computing **complexity maps** under end-to-end evaluation. Peak GPU memory reflects the cost of computing complexity maps. **Red** indicates the best.

| Model | Peak GPU memory | HCP | | ADNI-AD | |
|---|---|---|---|---|---|
| | | ACC ↑ | F1 ↑ | ACC ↑ | F1 ↑ |
| Shannon Entropy | 3.58 GB | 78.43 | 77.94 | 75.00 | 72.73 |
| Laplacian Response | 0.27 GB | 77.40 | 78.20 | 68.40 | 67.54 |
| MSE | 0.69 GB | 78.37 | 77.71 | 73.96 | 70.97 |
| Intensity Variance | **0.27 GB** | **85.18** | **84.59** | **75.69** | **75.33** |

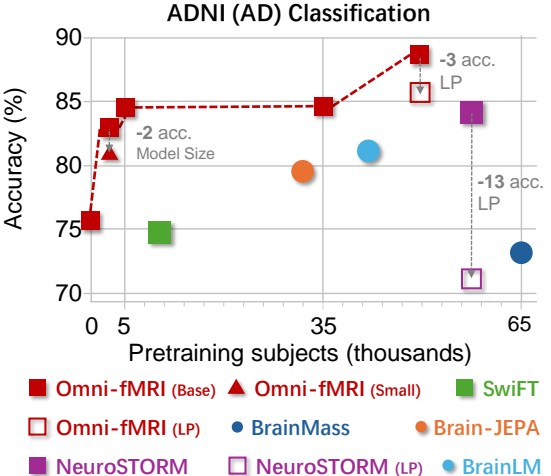

*Figure 4.* **Data Scaling and Linear Probing.** Classification performance on the ADNI (AD) dataset across different models and settings. Note that pretraining on zero subjects corresponds to end-to-end supervised training. LP: linear probing.

**Ablation of Patch Normalization** Contrary to standard MAE frameworks for natural images, we omit per-patch normalization in fMRI pretraining. While patch-wise normalization compensates for extrinsic variations such as illumination in images, fMRI voxel intensities—after global Z-score normalization—encode meaningful relative activation magnitudes. Applying normalization independently within each patch artificially equalizes local signal distributions, thereby disrupting the inherent signal-to-noise hierarchy across voxels and suppressing amplitude-based information critical for neural representation learning. As a result, removing patch normalization improves performance by up to 4 percentage points (Table 8); further analysis is provided in Appendix C.3.

*Table 8.* Ablation of **Patch-Normalization**. Red indicates the best.

| Model | HCP | | ABCD | | ADNI-AD | |
|---|---|---|---|---|---|---|
| | ACC↑ | F1↑ | ACC↑ | F1↑ | ACC↑ | F1↑ |
| With Patch-Norm | 86.61 | 85.11 | 69.17 | 69.16 | 80.21 | 77.94 |
| w.o. Patch-Norm | 87.05 | 85.57 | 70.21 | 69.96 | 84.72 | 84.33 |

**Ablation of Threshold** We evaluate the sensitivity of the patch subdivision threshold $\tau$ across two downstream tasks under an end-to-end supervised training protocol. Empirical results demonstrate that our selected value (0.25) yields a balance of performance and computational cost (Table 9).

*Table 9.* Ablation analysis of the patch subdivision **threshold**. *Token* denote the input sequence length to the encoder.

| Threshold | Token | ABCD | | HCP | |
|---|---|---|---|---|---|
| | | ACC↑ | F1↑ | ACC↑ | F1↑ |
| 0 | 14 K | | Out of Memory | | |
| 0.1 | 4.9 K | 67.29 | 67.28 | 85.77 | 85.56 |
| 0.2 | 4.6 K | 66.67 | 66.66 | 85.18 | 84.59 |
| 0.25 | 4.3 K | 67.29 | 67.24 | 85.10 | 84.66 |
| 0.3 | 3.9 K | 64.58 | 64.48 | 85.10 | 84.66 |

### 4.5. Interpretability

We examine whether the voxel-level representations learned by Omni-fMRI are neurobiologically meaningful by assessing their alignment with task-related regions. We utilize *Neurosynth* (Yarkoni et al., 2011), a large-scale meta-analytic platform, to generate consensus activation maps for specific terms (e.g., "Alzheimer"). To investigate the model's internal focus, we employ Integrated Gradients (IG) to compute voxel-wise attributions. By averaging the attribution maps across all subjects, we quantitatively evaluated the spatial correspondence with Neurosynth standards (Fig. 5). As evidenced by the similarity metrics, Omni-fMRI achieves superior alignment (SSIM: 0.5639, RMSE: 0.0562) compared to the baseline NeuroSTORM (SSIM: 0.4636, RMSE: 0.1386). These observations confirm that Omni-fMRI successfully captures interpretable, task-specific patterns that align with established pathophysiology.

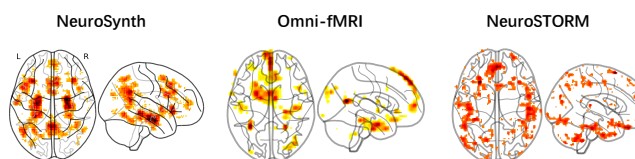

*Figure 5.* **Visualization of key brain regions associated with Alzheimer. Left:** Meta-analytic reference map from Neurosynth. **Middle:** Subject-averaged saliency map derived from Omni-fMRI using Integrated Gradients. **Right:** Corresponding saliency map from the NeuroSTORM baseline.

## 5. Discussion

The rapid growth of high-resolution and multimodal neuroimaging data has highlighted the limitations of atlas-based parcellation (Hayden et al., 2025). By discretizing the brain into fixed regions, such representations fail to capture the continuous nature of functional organization, complex interactions, and inter-subject variability. These issues are further amplified in foundation modeling, where task-agnostic generalization is essential, yet the choice of atlas inherently imposes task-dependent inductive biases on the learned representation (Wang et al., 2025b). This motivates an atlas-free fMRI formulation. In this work, we introduce Omni-fMRI as a principled solution to this dilemma. Our extensive empirical results demonstrate that it consistently outperforms ROI-based approaches across diverse downstream tasks, validating the representational advantage of preserving fine-grained spatial information. These findings suggest that voxel-level modeling is not merely a higher-resolution alternative, but a fundamentally more expressive representation for fMRI learning.

Despite its representational advantages, voxel-level modeling is severely constrained by the extreme dimensionality of fMRI data. Hierarchical approaches such as NeuroSTORM rely on rigid shifted-window designs that scale poorly: pretraining even a small backbone (7.7M parameters) takes roughly **13 days on 4 A6000 GPUs**, making larger-scale models practically infeasible. By comparison, ROI-based methods are computationally lighter but remain costly at scale; for instance, BrainMass (67.0M parameters) requires around **150 hours** of wall-clock time on **8 V100 GPUs** under a comparable pretraining setup. In contrast, Omni-fMRI employs dynamic patching to suppress functionally redundant regions, enabling efficient training at substantially larger scale. A 157M-parameter model can be pretrained for 35 epochs in only **32 hours on 4 NVIDIA A10G GPUs**. This improved scalability is not merely a practical advantage but a methodological enabler, allowing higher-capacity models to capture fine-grained functional geometry and long-range dependencies, which likely underpins the strong transfer performance observed across tasks.

**Limitations** The proposed dynamic patch tokenization relies on a heuristic spatiotemporal complexity measure based on variance thresholding, rather than a fully learned, end-to-end adaptive mechanism. While this choice provides robustness, interpretability, and computational efficiency, it may not optimally adapt to task-specific information distributions. An important direction for future work is to explore learned or differentiable patch selection strategies. Another limitation is that our current image retrieval evaluation relies on GLM-estimated stimulus responses. Thus, it does not fully assess end-to-end representation learning from raw continuous visual fMRI.

## Impact Statement

This work advances self-supervised representation learning for functional neuroimaging by enabling scalable, voxel-level foundation models without reliance on predefined atlases or region-based aggregation. By operating directly in voxel space and adaptively allocating computation through dynamic patching, our approach preserves fine-grained spatiotemporal information that is often discarded by ROI-based pipelines.

The proposed framework has the potential to benefit a wide range of downstream neuroimaging applications, including disease characterization, population-level brain modeling, and cross-dataset transfer learning. In particular, atlas-free voxel-wise representations may improve robustness to anatomical variability and reduce biases introduced by hand-crafted parcellations, which is especially relevant for heterogeneous clinical cohorts.

At the same time, this work is methodological in nature and does not introduce new clinical claims or decision-making systems. All data used for pretraining and downstream tasks are either open-access or licensed-access datasets. No sensitive personal information is disclosed.

Overall, we view this work as a step toward more flexible and faithful foundation models for neuroimaging, providing a technical basis for future research rather than an immediately deployable clinical tool.

## Acknowledgment

This work was supported by Brain Science and Brain-like Intelligence Technology - National Science and Technology Major Project (2021ZD0200500), the National Natural Science Foundation of China (62472206, 3254100307), National Key R&D Program of China (2025YFC3410000), Shenzhen Science and Technology Innovation Committee (RCYX20231211090405003, JCYJ20220818100213029), Guangdong Basic and Applied Basic Research Foundation (2026B1515020099), Guangdong S&T Program (Grant No. 2026B0101110003), Shanghai Municipal Special Program for Basic Research on General AI Foundation Models (2025SHZDZX026D05), Shenzhen Loop Area Institute under grant FPF10120250012, and the open research fund of the Guangdong Provincial Key Laboratory of Mathematical and Neural Dynamical Systems, the Center for Computational Science and Engineering at Southern University of Science and Technology, Shenzhen Key Laboratory of Smart Healthcare Engineering.

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

# A. Details of Materials

## A.1. Details of Datasets and Task

**UK Biobank (UKB).**   UKB is a large-scale population-based cohort comprising extensive phenotypic, genetic, and neuroimaging data collected from 38301 participants with a mean age of $68.37 \pm 7.48$ years at recruitment (Sudlow et al., 2015). Neuroimaging data were acquired using a standardized imaging protocol across multiple imaging centers, ensuring high consistency and data quality.

**Amsterdam Open MRI Collection (AOMIC).**   We employ two independent datasets from the AOMIC project for pretraining. PIOP1 comprises 168 subjects with a mean age of $22.15 \pm 1.83$ years (94 males, 67 females). PIOP2 comprises 180 subjects with a mean age of $21.99 \pm 1.83$ years (98 males, 81 females).

**Chinese Human Connectome Project (CHCP).**   To enhance the demographic diversity of our pre-training corpus, we incorporate the CHCP dataset, which consists of 244 subjects (117 males, 127 females). The CHCP cohort presents a significantly broader age distribution (18-79 years).

**Imaging Chinese Young Brains (ISYB).**   ISYB dataset includes 130 healthy Chinese Han young adults. Their age was in the range of 18 to 30 years, with a mean age of $22.52 \pm 2.61$ years. The sample consisted of 98 females and 32 males.

**Adolescent Brain Cognitive Development (ABCD).**   We incorporate the ABCD dataset, which includes 1680 participants aged 9–10 years as described in the original study (Casey et al., 2018). In contrast to UKB, this cohort focuses on early brain development, with participants recruited at 9–10 years of age.

**Autism Brain Imaging Data Exchange (ABIDE).**   ABIDE is a multi-site, open-access initiative that aggregates structural and resting-state fMRI scans - alongside rich phenotypic data - from individuals with autism spectrum disorder (ASD) and matched typically developing controls to accelerate reproducible neuroimaging research (Di Martino et al., 2014). The dataset comprises 609 participants, including individuals with ASD and age-matched typical controls ranging from 6 to 58 years of age.

**Human Connectome Project (HCP).**   We utilize the HCP dataset as our primary benchmark for pre-training and initial evaluation, comprising 606 subjects. This cohort represents a young adult population with mean age of $28.79 \pm 3.53$ years. The dataset provides high-quality, standardized data ideal for establishing baseline model performance.

**Parkinson's Progression Markers Initiative (PPMI).**   PPMI is a global, longitudinal, observational study aimed at identifying biomarkers of Parkinson's disease (PD) progression (Marek et al., 2011). For our disease classification task, we utilized a curated subset of the PPMI cohort; after quality control and task-specific filtering, 331 subjects with complete resting-state fMRI and diagnostic annotations were retained (age range: 35–84 years), covering prodromal PD, clinically diagnosed PD, and healthy control groups.

**Alzheimer's Disease Neuroimaging Initiative (ADNI).**   The Alzheimer's Disease Neuroimaging Initiative (ADNI) is a large-scale, longitudinal, observational study designed to characterize the progression of Alzheimer's disease (Jack Jr et al., 2008). For our disease classification task, we utilized a curated subset of the ADNI cohort; after quality control and task-specific filtering, 497 subjects with complete resting-state fMRI and diagnostic annotations were retained (age range: 55–90 years), spanning cognitively normal, mild cognitive impairment (MCI), and Alzheimer's disease (AD) groups.

**The Southwest University Adult Lifespan Dataset (SALD).**   SALD is a comprehensive, open-access neuroimaging resource (Wei et al., 2017). We utilized rs-fMRI data from 493 subjects to perform an age regression task, leveraging the dataset's broad and continuous age distribution.

**The Brazilian High-Risk Cohort (BHRC).**   BHRC is an extensive, ongoing longitudinal study dedicated to characterizing the developmental trajectories of psychopathology in youth (de Oliveira et al., 2024). We utilized preprocessed BHRC data acquired from the Reproducible Brain Corpus (RBC) (Shafiei et al., 2025). From this dataset, a cohort of 465 subjects was subsequently employed to perform binary sex classification.

**Nathan Kline Institute (NKI).** NKI is a comprehensive, open-science lifespan dataset designed to characterize the complex relationships between brain connectivity and behavioral phenotypes within a representative community cohort (Telesford et al., 2023). For our tasks, preprocessed data were obtained via the RBC (Shafiei et al., 2025). Leveraging the dataset's broad age distribution and granular educational metadata, we performed age-based regression across a sample of 717 participants and educational stratification with 331 participants. Specifically, subjects were categorized into three distinct groups based on educational attainment: primary education (Grades 1–6), secondary education (Grades 7–12), and higher education (those with graduate-level degrees).

**Natural Scenes Dataset (NSD)** We evaluate our framework on the Natural Scenes Dataset (NSD), aiming to map neural activity to visual semantics with high precision. To ensure high-fidelity neural representations, we utilize single-trial GLM betas (specifically the `betas_fithrf_GLMdenoise_RR` version) as model inputs. These estimates, optimized via HRF fitting and GLMdenoise, provide whole-brain voxel responses at a 2 mm isotropic resolution. This high-resolution, high-SNR input allows the model to leverage fine-grained cortical activity patterns for decoding. Our experiments are conducted under a strict *within-subject* setting, where independent models are trained for each subject. To rigorously assess generalization to unseen images, we employ a challenging split strategy: models are trained exclusively on the 515 shared images (viewed by all subjects). The evaluation is performed on an extensive scale across 6 subjects (Subj01–Subj06), utilizing all remaining unique images viewed by each subject. These test images are drawn from the full 73k NSD corpus, amounting to approximately 9,000 images per subject. This accumulates to a total of $\sim$54,000 unique test trials ($6 \times 9,000$), ensuring a statistically robust assessment of the model's performance over a highly diverse distribution of visual semantics. Decoding performance is quantified using a standard 100-way image retrieval task. For each test query, we construct a candidate pool consisting of the ground-truth image and 99 distractors that are randomly sampled from the test set. This setup simulates a realistic retrieval scenario, requiring the model to discriminate the target visual content from random interference. We rank the candidates based on the cosine similarity between the predicted brain embedding and the CLIP embeddings in the pool. We report Top-1, Top-5, and Top-10 accuracy to comprehensively measure the retrieval precision.

**HCP TASK.** The task-based fMRI dataset from the Human Connectome Project (HCP) encompasses a comprehensive battery of seven cognitive domains, stratified into 23 distinct experimental conditions. Working Memory evaluates executive function using an n-back paradigm, employing a factorial design that intersects memory load (0-back, 2-back) with four stimulus categories (faces, bodies, tools, and places) to yield eight specific contrasts. The Motor task performs somatotopic mapping via visual cue-induced movements across five effectors: left/right foot, left/right hand, and tongue. Emotion Processing assesses affective reactivity by contrasting blocks of fearful versus neutral faces. The Gambling task investigates reward circuitry and decision-making under risk, differentiated by win and loss outcomes. Language Processing targets semantic and phonological processing, delineating arithmetic calculation (math) from auditory narrative comprehension (story). Relational Processing tests analogical reasoning, requiring participants to identify dimensional matches (relation) versus object identity (match). Finally, Social Cognition probes Theory of Mind (ToM) mechanisms, where participants interpret geometric interactions as either socially meaningful (mental) or lacking social intent (random).

**Forrest Gump Audio Movie fMRI Dataset.** StudyForrest dataset comprises 20 subjects with a mean age of 26.6 years and an age range of 21–38 years (12 males, 8 females). The data includes 7T fMRI recordings during a 2-hour auditory movie presentation, with a total of 3,599 volumes per subject.

## A.2. Details of Baseline Models

In this section, we introduce our baseline models.

**BrainGNN** is a graph neural network architecture designed to analyze brain connectomes at the population level by learning interpretable node- and region-specific biomarkers. For each subject, it constructs a graph whose nodes correspond to brain regions and whose edges encode statistical dependencies between regional fMRI time series. The functional connectivity (FC) matrix is used as the node-level feature representation, and subject-level labels provide supervision for learning anatomically meaningful subgraphs and node embeddings. Partial correlations between fMRI time series are computed and assigned as edge attributes, enabling the model to capture conditional dependencies between brain regions rather than only marginal pairwise correlations. During training, we use batch size of 64, learning rate of 0.001, and weight decay of 0.5.

**TFF** (Malkiel et al., 2021) is a Transformer-based framework designed for fMRI analysis. It employs a two-phase training strategy: first, a self-supervised pre-training phase where the model learns to reconstruct 3D volumes from a collection of

fMRI scans; and second, a fine-tuning phase where the pre-trained model is optimized for specific downstream tasks using ground truth labels. We use default settings from the repository.

**SwiFT (Swin 4D fMRI Transformer)** addresses the challenge of modeling high-dimensional spatiotemporal brain dynamics by learning directly from raw 4D fMRI volumes, thereby avoiding the information loss introduced by hand-crafted features. Through its 4D windowed attention mechanism and computationally efficient architecture, SwiFT outperforms recent state-of-the-art models on large-scale datasets. Moreover, contrastive self-supervised pretraining further enhances its downstream performance, highlighting SwiFT's effectiveness as an end-to-end framework for functional brain imaging. In our experiments, we fine-tune SwiFT with a batch size of 16, a learning rate of $1 \times 10^{-6}$, a weight decay of 0.01, and a training duration of 30 epochs.

**BrainLM** is the first fMRI foundation model specifically designed to capture the spatiotemporal dynamics of brain activity through a Transformer-based masked autoencoder architecture. It employs the AAL-424 atlas for the parcellation of brain regions, transforming fMRI recordings into a 424-dimensional set of ROIs. The model is trained on an extensive dataset consisting of 6,700 hours of fMRI data derived from 77,298 recordings across the UK Biobank and the Human Connectome Project. For downstream fine-tuning, we choose the following hyperparameters: batch size of 64, learning rate of 0.00001, and training duration of 30 epochs.

**BrainMASS** utilizes the Schaefer 100-region atlas for the parcellation of brain networks. It was trained on 26 datasets, encompassing a total of 64,584 subjects, including data from the UK Biobank (UKB), Human Connectome Project (HCP), and ADHD-200, etc. The model is designed to learn representations of functional brain networks by integrating both masked ROI modeling and latent representation alignment techniques, thereby enhancing its diagnostic performance. The following hyperparameters were selected for downstream fine-tune: a batch size of 64, a learning rate of 0.0002, and a training duration of 100 epochs.

**Brain-JEPA** employs an advanced parcellation strategy that combines the Schaefer-400 cortical regions with the Tian-Scale III subcortical regions, resulting in a total of 450 ROIs. The model is trained on 80 % of the UK Biobank (UKB) data and utilizes a joint-embedding predictive architecture. This architecture incorporates both spatial and temporal masking techniques to capture dynamic patterns of brain activity. The model is trained with the following hyperparameters: a batch size of 16, a learning rate of $4 \times 10^{-4}$, over a total of 50 epochs.

**NeuroSTORM** establishes a general-purpose neuroimaging foundation model that learns directly from raw 4D fMRI volumes, avoiding the information loss associated with projecting data onto pre-defined atlases or connectomes. It employs a Shifted-Window Mamba (SWM) backbone. The model is pre-trained on a large-scale fMRI data from over 50,000 subjects—spanning the UK Biobank, ABCD, and HCP datasets. The model utilizes a Spatiotemporal Redundancy Dropout (STRD) module to mitigate data redundancy while capturing long-range dependencies.

### A.3. Configuration

The pre-training phase was conducted using four NVIDIA A10G GPUs for a duration of approximately 32 hours. For downstream evaluation, we transitioned to an NVIDIA A800 GPU to perform the inference tasks (Table 10).

*Table 10.* Hyperparameter settings of our model. (BS: Batch size)

| config | value |
| --- | --- |
| **Pre-train configs** | |
| optimizer | AdamW |
| optimizer momentum | $\beta_1, \beta_2 = 0.9, 0.95$ |
| learning rate schedule | warmup cosine schedule |
| learning rate | $2 \times 10^{-4}$ |
| minimal learning rate | $1 \times 10^{-6}$ |
| weight decay | 0.05 |
| warmup epochs | 5 |
| total batch size | $4\,GPUs \times 6$ BS = 24 |
| training epochs | 35 |
| encoder depths | 12 |
| number of attention heads | 12 |
| embedding dimension | 768 |
| decoder depths | 8 |
| decoder numbers of attention heads | 16 |
| decoder embedding dimension | 512 |
| mask ratio | 0.75 |
| threshold $\tau$ | 0.25 |
| patch scales $K$ | 2 |
| base patch size | (4, 4, 4) |
| Background threshold | $1 \times 10^{-3}$ |
| **Downstream configs** | |
| optimizer | AdamW |
| optimizer momentum | $\beta_1, \beta_2 = 0.9, 0.999$ |
| learning rate schedule | linear warmup and decay |
| learning rate | $1 \times 10^{-5}$ |
| head learning rate | $1 \times 10^{-4}$ |
| weight decay | 0.05 |
| layer decay | 0.75 |
| total batch size | $2\,GPUs \times 16$ BS = 32 |
| warmup epochs | 2 |
| training epochs | 30 |

# B. Details of Downstream Tasks

### B.1. Details of Image Retrieval

We evaluate our framework on NSD, utilizing single-trial GLM betas as inputs to represent whole-brain fMRI responses at 2mm isotropic resolution. Models are trained in a strict within-subject setting using only the 515 shared images. To strictly assess generalization, our evaluation is conducted on an extensive scale across 6 subjects (Subj01–Subj06). The test set comprises all remaining unique images viewed by each subject—approximately 9,000 images per subject drawn from COCO datasets (Lin et al., 2014). This accumulates to a total of ~54,000 unique test trials (6 × 9,000), ensuring a statistically robust assessment over a highly diverse distribution of visual semantics.

Decoding performance is quantified using a standard 100-way image retrieval task. For each test query, we construct a candidate pool consisting of the ground-truth image and 99 distractors randomly sampled from the test set. We rank the candidates based on the cosine similarity between the predicted brain embedding and the CLIP embeddings in the pool. We report Top-1, Top-5, and Top-10 accuracy to measure the retrieval precision.

## B.2. Details of Emotion Prediction

Emotion prediction was conducted on the StudyForrest dataset (Hanke et al., 2016), which provides continuous fMRI recordings acquired while participants watched the full-length movie *Forrest Gump*, together with time-resolved emotion annotations. The data were preprocessed using DeepPrep and normalized to MNI space. The fMRI time series were segmented into temporal blocks of 40 frames (TR = 0.72 s, input size $96 \times 96 \times 96 \times 40$), and each block was labeled using the averaged emotion scores. The original dataset contains six emotion dimensions (happiness, surprise, fear, sadness, anger, and disgust). After removing zero values, the remaining four emotions were sparsely represented, whereas happiness and sadness occurred most frequently; therefore, emotion prediction was formulated as a regression task focusing exclusively on these two emotion dimensions (Fig. 6). For each experiment, the data were randomly split into training, validation, and test sets with a ratio of 7:1:2, and the entire procedure was repeated across three independent random splits to ensure robustness. For benchmarking, we compared our method with recent voxel-level models, including NeuroSTORM and SwiFT, using the same data splits and preprocessing pipeline.

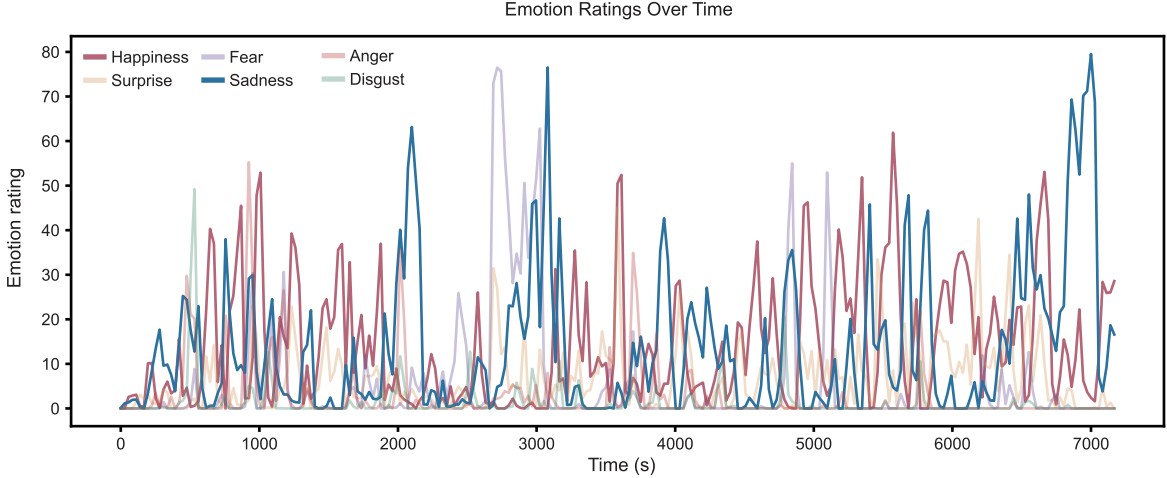

*Figure 6.* Temporal emotion labels used for regression in the StudyForrest dataset.

## B.3. Details of Brain State Prediction

We evaluated the model's performance on brain state prediction using the HCP Task dataset. The fMRI time series were segmented into task activation blocks based on experimental triggers and formulated as a rigorous 23-way classification problem. To ensure robustness and fair comparison, experiments were conducted across three random data splits, with identical data partitions applied to all models. We comprehensively assessed data efficiency by benchmarking Omni-fMRI against state-of-the-art baselines (BrainMASS, SwiFT, TFF, BrainGNN, and NeuroSTORM) under full supervision (100

As presented in Table 11, our method achieves state-of-the-art performance across all data regimes. Under the full supervision setting (100%), Omni-fMRI attains a leading accuracy of 50.17%, outperforming the strongest baseline, BrainMASS (49.85%). However, the superiority of our framework becomes most pronounced in data-scarce scenarios. In the 50% data setting, while BrainMASS performance drops significantly to 39.68%, Omni-fMRI maintains a high accuracy of 46.19%.

Most notably, in the extreme few-shot setting (10%), Omni-fMRI demonstrates exceptional resilience, sustaining an accuracy of 45.29%. This represents a minimal performance decay of less than 5% despite a 90% reduction in training data. In stark contrast, baseline models suffer catastrophic degradation: BrainMASS plummets to 23.12%, and BrainGNN fails to capture meaningful patterns, dropping to 7.38%. These results statistically validate that Omni-fMRI learns robust, generalizable functional representations that require significantly fewer samples to fine-tune effectively, far surpassing the data efficiency of existing voxel-level and graph-based approaches.

*Table 11.* Comparison of HCP Task Performance (Accuracy %) averaged over three runs. "Full", "50%", and "10%" denote the percentage of training data used (few-shot settings). * denotes a large effect size with Cohen's d $\geq 0.8$.

| Model | Full (100%) | Few-shot (50%) | Few-shot (10%) |
|---|---|---|---|
| BrainGNN | $19.97 \pm 1.63$ | $19.22 \pm 2.19$ | $7.38 \pm 1.20$ |
| BrainMASS | $49.85 \pm 0.61$ | $39.68 \pm 1.07$ | $23.12 \pm 0.53$ |
| TFF | $23.79 \pm 0.75$ | $23.50 \pm 1.29$ | $19.76 \pm 6.01$ |
| SwiFT | $22.72 \pm 1.63$ | $24.41 \pm 0.24$ | $23.17 \pm 1.02$ |
| NeuroSTORM | $25.45 \pm 1.48$ | $21.33 \pm 0.94$ | $20.00 \pm 2.04$ |
| Omni-fMRI | **$50.17 \pm 1.64$** | **$46.19 \pm 0.61$ *** | **$45.29 \pm 1.09$*** |

## C. Detailed Ablations

### C.1. Scaling Law

We conducted a scaling analysis by pretraining the Base model on four different dataset sizes, and compared two model sizes. The pretrained models were evaluated on downstream classification tasks (Table 12 and Table 13). For the pretraining datasets, we split 10% to validate.

*Table 12.* Model size scaling using a combined dataset of 3,008 sessions from HCP, CHCP, ABCD, PIOP1, and PIOP2

| Model | Size | HCP | | ABCD | | ADNI-AD | |
|---|---|---|---|---|---|---|---|
| | | ACC | F1 | ACC | F1 | ACC | F1 |
| Small | 47M | 85.18 | 84.63 | 69.37 | 69.16 | 80.90 | 79.11 |
| Base | 157M | **89.29** | **88.12** | **70.21** | **70.20** | **82.99** | **81.79** |

*Table 13.* Data-size Scaling

| Pretraining Datasizes | HCP | | ABCD | | ADNI-AD | | ADNI-MCI | |
|---|---|---|---|---|---|---|---|---|
| | ACC | F1 | ACC | F1 | ACC | F1 | ACC | F1 |
| 0 (End to End) | 85.18 | 84.59 | 65.62 | 65.53 | 75.69 | 75.33 | 60.62 | 58.35 |
| 3,008 | 89.29 | 88.12 | 70.21 | 70.20 | 82.99 | 81.79 | 65.62 | 66.20 |
| 5,000 | 87.05 | 85.57 | 70.21 | 69.96 | 84.72 | 84.33 | 65.42 | 61.82 |
| 34,471 | 91.07 | 90.13 | 72.71 | 72.71 | 84.72 | 84.21 | 72.50 | 72.10 |
| 49,497 | **95.54** | **95.05** | **77.01** | **76.91** | **88.77** | **88.60** | **72.92** | **72.87** |

### C.2. Ablation of Patch Partition

**Local Shannon Entropy** First compute the temporally averaged volume $\bar{I}$ to obtain a stable anatomical representation. Local Shannon entropy is then computed to quantify the texture randomness and information diversity of $\bar{I}$. The metric is derived from the local intensity histogram constructed within each 3D patch. Let $p(k)$ denote the probability mass of the $k$-th intensity bin within a patch $\mathcal{P}$ of $\bar{I}$, computed over $B$ bins (where $B = 512$). The entropy-based complexity score $S_{Ent}$ is defined as:

$$S_{Ent}(\mathcal{P}) = -\sum_{k=1}^{B} p(k) \log_2 (p(k) + \epsilon), \tag{5}$$

where $\epsilon$ is a small constant for numerical stability. This formulation highlights regions with diverse signal distributions rather than simple magnitude changes.

**Laplacian Response** We first obtain the temporally averaged volume $\bar{I}$ to mitigate functional noise. This volume is convolved with a discrete 3D Laplacian kernel $\mathbf{K}_\Delta$ using a 26-connectivity neighborhood (center weight $-26$) to

approximate second-order derivatives. The complexity score is obtained by aggregating the absolute response magnitude via the expectation operator $\mathbb{E}_{\mathcal{P}}$ (implemented as 3D average pooling) over the local patch $\mathcal{P}$:

$$S_{Lap}(\mathcal{P}) = \mathbb{E}_{\mathbf{x} \in \mathcal{P}} \left[ \left| (\bar{I} * \mathbf{K}_{\Delta})(\mathbf{x}) \right| \right], \tag{6}$$

where $*$ denotes the 3D convolution operator and $\bar{I}$ represents the time-averaged intensity volume.

**Reconstruction Error (MSE)**   A signal compressibility proxy is employed, premised on the observation that fine-grained details are lost during low-resolution representation. A reconstructed volume $\hat{I}$ is generated via a downsampling operation $\mathcal{D}(\cdot)$ followed by an upsampling operation $\mathcal{U}(\cdot)$ using trilinear interpolation:

$$\hat{I} = \mathcal{U}\left( \mathcal{D}(I; s); s \right), \tag{7}$$

where $s$ denotes the scaling factor. The complexity map is defined as the Mean Squared Error (MSE) between the original and reconstructed volumes. Using the notation $\mathbb{E}_{\mathcal{P}}$ for local averaging, the score is expressed as:

$$S_{MSE}(\mathcal{P}) = \mathbb{E}_{\mathcal{P}}\left[ \left( I - \hat{I} \right)^2 \right]. \tag{8}$$

High reconstruction error indicates regions containing high-frequency spectral components that necessitate finer tokenization resolution.

## C.3. Ablation of Patch-Normalization

In standard MAE frameworks developed for natural images, applying per-patch normalization before loss computation is a widely adopted practice. Theoretically, this operation enhances robustness by rendering the reconstruction target invariant to local contrast changes, forcing the model to focus on structural semantics rather than low-level intensity statistics.

However, our empirical results present a counter-intuitive finding: removing Patch-Norm consistently outperforms the standard normalized counterpart on 4D fMRI data. We attribute this discrepancy to the fundamental difference between optical imaging and functional neuroimaging, particularly concerning the interaction with global preprocessing strategies: Global Z-scoring.

**Semantic Meaning of Intensity.**   In natural images, absolute intensity often represents extrinsic factors that are irrelevant to the object identity. In contrast, for fMRI data, voxel intensity carries intrinsic biological semantics—specifically, the amplitude of the BOLD signal correlates with the strength of neural activation. Global Z-scoring standardizes the baseline across the entire scan, meaning that high-magnitude values indicate significant neural events, while low-magnitude values typically correspond to background noise or resting states. By applying Patch-Norm, we inadvertently discard this relative amplitude information. A patch containing a strong neural activation is normalized to the same statistical distribution as a patch containing mostly background noise. Consequently, the loss function treats signal-rich regions and noise-dominated regions with equal importance, potentially hindering the model from prioritizing biologically meaningful patterns.

**Noise Amplification in Low-Signal Regions.**   Furthermore, fMRI data is inherently noisy with a varying SNR across spatial regions. Without Patch-Norm, the reconstruction loss is naturally weighted by the signal magnitude; the model penalizes errors in high-activation regions more heavily than in flat background regions. Patch-Norm, by enforcing unit variance locally, acts as a contrast stretching operation. In signal-void regions (where the local variance is dominated by thermal noise), this operation significantly amplifies the noise floor. The decoder is then forced to allocate capacity to reconstruct these amplified noise patterns, distracting it from learning robust spatiotemporal representations of brain dynamics.

A comparative analysis based on 5,000 UKB subjects reveals that models trained without patch normalization outperform their normalized counterparts across three distinct downstream tasks.

## C.4. Self-supervised proxy tasks

Our ablation results (see Tab. 14) demonstrate a counter-intuitive finding compared to recent trends in computer vision: the reconstruction-based objective (MAE) consistently outperforms the JEPA for voxel-level fMRI pretraining.

**Sensitivity to Noise Magnitude.** According to Van Assel et al. (2025), reconstruction-based methods are theoretically preferable when the magnitude of irrelevant noise features is low relative to the informative signal variance. In standard fMRI pipeline, voxel intensities undergo rigorous preprocessing and global Z-scoring, which standardizes the baseline noise floor. Consequently, the high-variance components in the data typically correspond to biologically meaningful BOLD signal fluctuations rather than irrelevant background noise. Under this *low-noise regime*, the reconstruction objective naturally prioritizes capturing these high-variance informative components without the need for complex regularization. Furthermore, our proposed adaptive tokenization strategy—utilizing dynamic patch sizing and background removal—explicitly filters out information-sparse regions, thereby ensuring that the model focuses exclusively on reconstructing high-informative biological signals. Conversely, joint-embedding methods excel in high-noise regimes where the signal is obscured by high-magnitude irrelevant features that must be filtered out via aggressive augmentation—a scenario that is effectively mitigated by both our preprocessing and tokenization mechanisms.

**The Augmentation Alignment Dilemma.** A critical constraint of JEPA is their heavy reliance on data augmentations. Theoretical analysis shows that to achieve asymptotic optimality, joint-embedding methods require the augmentation distribution to be strictly aligned with the irrelevant noise features in the data. In natural image domains, this is achievable via color jittering or blurring (simulating illumination/sensor noise). However, for 4D fMRI volumes, defining "semantic-preserving" augmentations is notoriously difficult. Arbitrary intensity transformations risk destroying the amplitude information intrinsic to the BOLD signal, while spatial deformations usually misalign anatomical correspondences. When "prior information on effective augmentations is scarce", reconstruction-based approaches are the robust choice. MAE bypasses this dependency by operating directly in the input space, effectively learning the spatiotemporal dynamics of brain activity without requiring handcrafted invariances that may not hold for neuroimaging data.

**Variance as Information.** Finally, the theoretical bias of reconstruction methods towards "variance-explaining features", often considered a limitation in computer vision, proves advantageous for fMRI. In functional neuroimaging, variance over time is the primary marker of neural information processing. By minimizing reconstruction error, MAE is implicitly guided to encode the most active and dynamic brain regions, thereby aligning the pretraining objective with the downstream goal of detecting neural activation patterns.

*Table 14.* Ablation of Self-Supervised Tasks

| Model | HCP | | ABCD | | ADNI-AD | | ADNI-MCI | |
|---|---|---|---|---|---|---|---|---|
| | ACC | F1 | ACC | F1 | ACC | F1 | ACC | F1 |
| JEPA | 87.69 | 87.62 | 68.96 | 68.36 | 82.99 | 82.81 | 64.17 | 57.65 |
| MAE | 89.29 | 88.12 | 70.21 | 70.20 | 82.99 | 81.79 | 65.62 | 66.20 |

## C.5. Ablation of Mask Ratio

We conducted an ablation study on two masking ratios using a pretraining dataset of about 3,008 subjects. Results across four tasks show that a 0.9 masking ratio is counterproductive. Unlike redundant natural videos, fMRI signals are inherently informative, with their density further enhanced by our adaptive tokenizer. Consequently, 90% masking leaves insufficient context to distill robust latent features, leading to performance inferior to our chosen 0.75 mask ratio (Table 15).

*Table 15.* Ablation of Mask Ratio

| Mask Ratio | HCP | | ABCD | | ADNI-AD | | ADNI-MCI | |
|---|---|---|---|---|---|---|---|---|
| | ACC | F1 | ACC | F1 | ACC | F1 | ACC | F1 |
| 0.9 | 82.59 | 80.66 | 65.83 | 65.62 | 81.25 | 80.09 | 63.96 | 63.45 |
| 0.75 | 89.29 | 88.12 | 70.21 | 70.20 | 82.99 | 81.79 | 65.62 | 66.20 |

